# PrEP Use Awareness and Interest Cascade among MSM and Transgender Women Living in Bali, Indonesia

**DOI:** 10.3390/tropicalmed5040158

**Published:** 2020-10-10

**Authors:** Rissa Cempaka, Brigitta Wardhani, Anak Agung Sagung Sawitri, Pande Putu Januraga, Benjamin Bavinton

**Affiliations:** 1Magister Program in Public Health, Faculty of Medicine, Udayana University, Denpasar, Bali 80361, Indonesia; ayurissacempaka@gmail.com; 2Center for Public Health Innovation, Faculty of Medicine, Udayana University, Denpasar, Bali 80361, Indonesia; gittadhyah@gmail.com; 3Department of Public Health and Preventive Medicine, Faculty of Medicine, Udayana University, Denpasar, Bali 80361, Indonesia; sagungsawitri@gmail.com; 4The Kirby Institute, University of New South Wales, Kensington, NSW 2052, Australia; bbavinton@kirby.unsw.edu.au

**Keywords:** PrEP, awareness, interest, cascade, MSM, transgender women, Indonesia

## Abstract

Indonesia has not implemented HIV pre-exposure prophylaxis (PrEP) despite global calls for its scale-up, and there is limited information about attitudes towards PrEP among its potential users. We aim to present a PrEP cascade among men who have sex with men (MSM) and transgender women (known locally as “waria”) in Denpasar, Bali, from a cross-sectional survey with 220 HIV-negative MSM/waria recruited from one clinic in Denpasar. Only 16.4% of participants had heard of PrEP before. From first-to-last steps included in the cascade, we found 77.3% (170/220) of participants were classified with HIV high risk, 75.9% (129/170) perceived themselves as being at high risk, 81.4% (105/129) expressed interest in using PrEP, 78.1% (82/105) were willing to do PrEP procedures, 48.8% (40/82) were willing to pay 500,000–600,000 IDR, and only two participants had ever been on PrEP before (5.0% of those willing to pay and 0.9% of the total sample). Multivariate logistic regression analysis showed that self-perception of high HIV risk was lower among older age groups (*p* < 0.001 among 30–39; *p* = 0.002 among > 40) and higher among participants with multiple sex partners (*p* = 0.016). Interest in using PrEP was lower among participants with high social engagement as MSM/waria (*p* = 0.002) and was higher among participants with multiple sex partners (*p* = 0.020) and inconsistent condom use (*p* = 0.011). This study has shown a significantly low level of PrEP awareness among its participants and decreases in interest in PrEP use due to procedure and cost. It suggested that an appropriate PrEP campaign is needed if PrEP is going to be introduced in Indonesia.

## 1. Introduction

In 2018, UNAIDS reported that there were about 640,000 people living with HIV (PLHIV) in Indonesia [1], with the prevalence of 0.4% among the general population [1]. Based on the integrated HIV bio-behavioral surveillance (IBBS) report in 2018–2019, national HIV prevalence was 17.9%, 13.6%, 11.9%, and 2,1% among men who have sex with men (MSM), people who inject drugs (PWID), transgender women/TGW (known locally as “waria”), and female sex workers, respectively [2]. In Denpasar, Bali, Indonesia, the 2019 IBBS report showed that HIV prevalence among MSM was 38.1% [2], while the 2015 IBBS report showed that HIV prevalence among female (direct) sex workers was 4.8%, female (indirect) sex workers was 5.6%, MSM was 36.0%, and incarcerated people 3.8% [3]. In view of the high prevalence of HIV among MSM and waria compared to the general population, addressing HIV prevention in these key populations could result in significant outcomes [4], such as a reduction in new HIV cases [5].

Consisting of some of the same active regiments used in antiretroviral (ARV) therapy (ART), HIV pre-exposure prophylaxis (PrEP) has been shown to protect against HIV infection in MSM and TGW [6,7], provided that users are adherent [8,9,10,11,12]. Individuals need to take PrEP during periods of high risk of exposure to HIV, a concept that has been termed “prevention-effective adherence” [12]. Although a plan to establish a pilot study on PrEP implementation for MSM population has been stated in its National Strategy and Action Plan 2015–2019 for HIV and AIDS prevention [13], Indonesia has not implemented PrEP and has fallen behind neighboring countries such as Thailand [14], where it has been integrated into its universal health coverage since 2018 [15] and resulted in an estimate of 16,000–17,000 PrEP users in Thailand as of 9 July 2020 [16]. Although literature suggesting direct PrEP contribution towards HIV epidemic in Thailand is not yet readily available, the fact that HIV prevalence has slowed down in recent years in this country should not be overlooked [17].

In Indonesia, none of the three PrEP regimens suggested by WHO [18] have been licensed, and PrEP is not accessible through the national health system [19]. Distribution of ARV as PrEP regulation is also not yet available. However, at the moment in Indonesia, PrEP can be bought online without prescription [20,21], despite the fact that ARV as used in PrEP can only be obtained through prescription [19]. Obtaining PrEP in Indonesia at the moment is also costly. For example, a one-month supply (30 pills) of PrEP can be bought without prescription for 1–2 million IDR (about 67–135 USD) [20,21], a price that is unaffordable to most. One option is to legally procure PrEP by acquiring a prescription locally and buying the pills from countries where PrEP use has been licensed, such as Thailand. However, the emerging PrEP-related cost could highly likely prohibit most potential PrEP users to access PrEP abroad, as they must consider the fly-out transportation cost, in-country accommodation and meal cost, as well as in-country PrEP-related costs [22,23,24,25].

Given the current HIV epidemic among MSM/waria in Indonesia [1,2,3], the inclusion of PrEP within Indonesia’s HIV response [24,25] should be prioritized. Regardless of the lack of regulatory, approval, or national guidelines for PrEP, comprehensive and focused studies and preparation to support PrEP best practice and legal distribution should be established so that enablers and barriers of successful PrEP implementation can be identified and addressed [9,10,26]. Data on how the potential users would respond towards PrEP are needed before the official introduction of PrEP in Indonesia. One option to present such data is using cascade analysis. Based on the steps included, cascade analyses can identify gaps in HIV prevention programs and, thus, help programs be more effective [27,28,29]. This analysis aimed to develop potential new PrEP cascades relevant to the specific stage of PrEP knowledge and rollout in Indonesia and determine factors associated with key steps in the cascade.

## 2. Methods

### 2.1. Data and Samples

We conducted a cross-sectional survey on HIV risk, PrEP awareness, and interest in PrEP among MSM/waria attending a non-government HIV testing and treatment clinic in Denpasar, Bali, Indonesia. The clinic is a very reputable clinic in its specialized services regarding sexually transmitted infections (STI) and HIV for the HIV key populations. We recruited 220 participants from August 2017 to April 2018. Included participants received a negative HIV test result during their visit, were 18 years or older, could participate in Indonesian language, and provided written informed consent.

### 2.2. Procedures

All eligible participants were invited to participate in the study by clinic staff. After receiving study information and giving consent, participants were asked to complete an interviewer-administered survey. All participants were interviewed face to face by the first author only; therefore, double participation could easily be avoided. Data were collected on a laptop using the SurveyGizmo online survey platform. The survey collected data on participants’ demographics, HIV risks and prevention-related knowledge, sex and HIV prevention-related behavior, PrEP awareness and knowledge, and interest in using PrEP. Regardless of participants prior knowledge of PrEP, all participants were provided with scripted information describing PrEP during the interview. Given that the interview involved this education about PrEP, we present two cascades in this analysis: cascade 1 includes the participants’ original awareness of PrEP as a cascade step, while cascade 2 excludes awareness as a cascade step. Compensation of 50,000 IDR (about 4 USD) for travel costs was provided to each study participant. 

### 2.3. Variables and Measures

The cascade steps and how they were determined is as follows:1Classified as high risk for HIV infection: measured by questions on sexual behavior and STI diagnoses; those who reported any STI diagnoses in the last 6 months, >1 sexual partner in the last 6 months, and/or reported condom-less anal intercourse (CLAI) with MSM/waria partners in the last 6 months were classified as high risk.2Self-perceived high risk for HIV infection: measured by the question, “Based on your sexual activities in the last 6 months, how likely do you perceive your risk of being infected with HIV?”; of 4 options provided for the participants (not at all at risk, less risky, risky, and highly risky) those answering “not at all” were classified as not perceiving themselves as high risk.3Aware of PrEP: measured by the question, “Have you ever heard of or received information about PrEP?”; those answering “yes” were classified as being aware of PrEP.4Interested in using PrEP: measured by the question, “Are you interested in using PrEP to protect yourself from HIV infection?”; those answering “yes” were classified as interested.5Willing to do PrEP procedures: measured by the question, “How willing are you to take PrEP if it involves regular consultation visits including voluntary counselling and testing (VCT) and liver function check?”; those answering “yes” were considered to be willing to do PrEP procedures.6Willing to pay 500,000 to 600,000 IDR for PrEP (the cost of locally available PrEP at the time): Measured by the question, “How willing are you to use PrEP if it costs 500,000–600,000 IDR per month?”; of the 5 options provided for the participants (highly unwilling, unwilling, slightly willing, willing, and highly willing) those answering “highly unwilling” and “unwilling” were classified as unwilling.7Already in PrEP: measured by the question, “Have you ever used ARV as PrEP before any sexual activities?”; those answering “yes” were classified as already on PrEP.

To be included in step 2 onward, participants had to meet the criteria for the previous steps. As mentioned above, all 7 steps were included in cascade 1, while step 3 was excluded from cascade 2.

Several covariates were derived from the survey questions. “Social engagement as MSM/waria” was measured from two questions regarding participants’ number of MSM/waria friends and time spent with MSM/waria friends. Participants’ answers to these questions were scaled and divided into two groups using the median as the cut-off; those with scores above the median were classified as having “high social engagement as MSM/waria”. “Adequate HIV risk/prevention knowledge” was determined from 8 questions which were used in the 2010 basic health research (*Riset Kesehatan Dasar, Riskesdas*) [30] held by the health ministry of Indonesia, such as “Can someone be infected with HIV from a mosquito bite?”. Participants’ answers to these questions were scaled and divided into two groups using the median as the cut-off; those with scores above the median were classified as having “adequate HIV risk/prevention knowledge”. “Consistent condom use” was derived from 4 questions on anal intercourse with MSM/waria partners, which included information on condom use and sexual position. Participants were classified as consistently using condoms if they answered, “always used condoms” to all questions.

### 2.4. Analysis

Analyses were conducted using Stata version 12 (Stata Corporation, College Station, Texas, USA). We present descriptive statistics of the sample, followed by the number and proportion of participants in each of the cascade steps. The denominator for the cascades were all participants in the sample. The number of participants in each step was determined, and percentages were calculated (proportion of the total sample; proportion of those included in the previous step).

We performed two bivariate and multivariate logistic regression models, based on cascade 2. The first model examined associations with self-perceived high HIV risk among those who were classified as at high risk for HIV infection. The second model examined associations with interest in PrEP use among participants classified as at high risk and who self-perceived their risk to be high. Statistically significant associations < 0.25 at the bivariate level were included in the multivariate models. We present odds ratios (OR), adjusted odds ratios (aOR), 95% confidence intervals (CI), and p-values for these associations.

### 2.5. Ethical Consideration

The protocol of primary data collection has obtained ethical approval from the Ethics Committee of Faculty of Medicine, Udayana University, Bali, Indonesia with No. 1745/UN.14.2/KEP/2017. Written informed consent were obtained from each participant.

## 3. Results

The participants ranged in age from 18 to 52 years, with a median age of 28 (IQR = 24–32); only 6.4% were aged over 40 years (Table 1). Most participants (90.9%) were not married to a woman. Under one-third (30.0%) was university-educated. Three-quarters were employed full-time, and 47.7% reported their income to be more than the Bali provincial minimum wage of 2.494 million IDR per month [31]. About one third (30.9%) were classified as having high social engagement as MSM/waria, and 42.3% of participants had adequate HIV risk/prevention knowledge. Regarding sexual behavior in the last 6 months, 33.2% participants had sex with women, 57.3% reported sex with one or more regular male partners, 58.2% reported sex with one or more casual male partners, and 15.5% reported having been paid for sex. The median number of sex partners was two (IQR = 1–3), ranging from one to 25 partners. In the last six months, half of the participants (50.5%) reported having > one sex partner, more than half (61.8%) participants reported inconsistent condom use, and 11.8% reported an STI diagnosis.

Thirty-six (16.4%) participants had heard of PrEP before participating in the study. Of these, 15 heard about PrEP from friends, 13 from the internet, eight from social media, seven from educational activities conducted at the recruiting clinic, and three from health workers. A total of 164 participants (74.5%) were interested in using PrEP (including participants who learned about PrEP for the first time from the interviewer). About half (n = 13) of the 24 participants who were not interested perceived that they were at minimal risk for HIV infection, nine were not interested due to the unwanted side effects, while two participants stated that condoms provided adequate protection from HIV. Only two participants had used PrEP in the last 6 months prior to the study: one procured PrEP with the help of his clinic, although it was not clear where he acquired the pills; the other received PrEP with the help of his overseas friend.

In the first PrEP cascade (as shown in Appendix A as a supplementary) where PrEP awareness was included, three-quarters (n = 170, 77.3%) were classified as at high HIV risk, and from these, 75.9% (n = 129) perceived themselves to be at high risk. Of those with high risk self-perception, only 17 participants (13.2% of the previous step) were aware of PrEP, representing an 86.8% decrease from the previous step.

In the second PrEP cascade (as shown in Figure 1 and Appendix A as a supplementary) where all participants were considered aware of PrEP, of the 129 with self-perceived high HIV risk, 81.4% (n = 105) were interested in using PrEP. Of those interested, 78.1% (n = 82) were willing to do the PrEP procedures, but only half (48.8%, n = 40) of those willing to do the PrEP procedures were willing to pay 500,000 to 600,000 IDR per month for it. Two participants, representing 5.0% of those willing to pay and only 0.9% of all participants, were already taking PrEP. There was a large percentage of decreases (that is, of > 10%) between each step of cascade 2.

In multivariate analysis, among those classified as having high HIV risk (n = 170), self-perception of high HIV risk (n = 129) was lower in older participants (for the 30–39 age group: aOR = 0.07, 95% CI = 0.03–0.29, *p* < 0.001; for the > 40 age group: aOR = 0.06, 95% CI = 0.01–0.34, *p* = 0.002) and was higher in those who reported multiple sex partners in the last six months (aOR = 3.08, 95% CI = 1.23–7.72, *p* = 0.016; Table 2). From the multicollinearity testing, we found that none of the independent variables was found to be highly correlated (r < 0.5).

In multivariate analysis, among those who were both classified as high risk and perceived themselves to be high risk (n = 129), interest in PrEP use (n = 105) was higher among those who have multiple sex partners (aOR = 4.26, 95% CI = 1.26–14.45, *p* = 0.020), higher among those with inconsistent condom use (aOR = 4.80, 95% CI = 1.43–16.09, *p* = 0.011), and lower among participants with high social engagement as MSM/waria (aOR = 0.17, 95% CI = 0.06–0.51, *p* = 0.002; Table 3). From the multicollinearity testing, we found that none of the independent variables was found to be highly correlated (r < 0.5).

## 4. Discussion

This study presented an alternative approach on how to conduct a PrEP use awareness and interest cascade analysis among the MSM/waria key population, specific to the Bali, Indonesia, setting. This study was cross-sectional rather than longitudinal, and the steps differed slightly from those commonly suggested by studies and WHO guidelines [8,32,33,34,35]. We found a very low level of PrEP awareness prior to the PrEP information given during the interview. However, after being provided with information, we found a high level of interest in PrEP use among those who perceived themselves as at risk. Self-perception of risk was associated with age and multiple sex partners, while interest in using PrEP was associated with social engagement as MSM/waria, multiple sex partners, and inconsistent condom use. Research on the various steps in the PrEP cascade is very limited in Indonesia, with only one other cross-sectional study PrEP awareness, knowledge, and willingness in PrEP use among MSM/waria in Bali, Indonesia [36].

An important early step on an individual’s PrEP use journey is the capacity to perceive oneself as at risk of HIV infection based on sexual behaviors. We found that three-quarters of those classified as at risk perceived themselves as such. Those over the age of 30 were less likely to perceive themselves to be at risk, in contrast to Thai research in MSM and transgender women showing that older participants were more likely to accurately self-assess their risk [37]. We also found that participants with multiple sex partners were more likely to perceive themselves as at risk (and in bivariate analysis, those with STI diagnoses), in line with other research [38]. A lack of self-perception of risk among those with potential HIV exposures can mean that those who need PrEP may not come forward for it. Furthermore, if clinicians are not skilled at taking sexual histories and assessing risk, patients who could benefit from PrEP may also be missed [39]. Efforts to educate the MSM/waria community about HIV risk should be strengthened. Further research on understanding the mismatch between risk and self-perception is warranted.

Awareness of PrEP is clearly a critical step to PrEP access and initiation and is often featured in PrEP cascade analyses [8,32,33,34]. Around the time of data collection period, there were limited PrEP campaigns in Indonesia, including in Bali. Therefore, we expected low PrEP awareness among participants. To address this, all participants were provided with scripted information describing PrEP during the interview. Unsurprisingly, in this study, we identified an exceptionally low level of PrEP knowledge in the sample. Only 16% of participants (36/220) had heard about PrEP prior to the study. Even in countries such as Ireland and USA where PrEP is legally available and accessible, PrEP awareness among MSM was only 34% and 16%, respectively [40,41]. Indeed, to be effective, PrEP programs require its potential users to be aware of its existence. Thus, a PrEP campaign is important if it is going to be implemented in Indonesia.

We found high interest in using PrEP, after participants had been informed about it. Among those classified as having high HIV risk and having self-perception of risk, lower social engagement as MSM/waria, having multiple sex partners, and inconsistent condom use were associated with interest in PrEP use. Our finding on social engagement is contrary to most studies of interest in PrEP, which typically find that higher engagement with other MSM/waria is strongly associated with higher interest [42,43,44]. This clearly warrants more investigation. One possible explanation could be that those with strong MSM/waria peer networks may be more likely to fear judgement or stigma from their peers about PrEP use or their sexual behavior, such as fear of rejection from partners (actual/potential), stigma of promiscuity and *chemsex* stereotypes labelling, and the fear of potential label stigma surrounding the PrEP medication and its users, as has been found in other setting [45]. Furthermore, there is low knowledge of PrEP and no norm of PrEP use, due to its lack of availability, MSM/waria are more likely to be ideologically committed to condom use. Thus, those with lower engagement with other MSM/waria may feel safer to express interest in using PrEP. When access to PrEP is scaled up in Indonesia, efforts will be needed to understand and address such factors.

In regard to HIV-related risky behaviors in the last 6 months, we found that among participants with high HIV risks and have perceived the risks, those with multiple MSM sex partners and inconsistent condom use were more interested in PrEP use as were reported by previous research globally [44,46]. The finding was expected that our participants with higher HIV risky behaviors were more interested in PrEP use. Moreover, if the 61.8% participants with inconsistent condom use kept on engaging in unsafe sex of CLAI, it could be challenging to keep the HIV status negative among them without any interventions. It indicates the potential cost-effectiveness of a PrEP program upon implementation.

Aside from lack of awareness and knowledge of PrEP, we identified that PrEP cost is likely to present major challenge to the use of PrEP in Indonesia. Interest in PrEP use decreased significantly when the average cost of one-month PrEP supply was introduced. Although PrEP regimens are in fact the same regimens used in ART as in HIV treatment [18], in Indonesia, HIV antiretroviral drug (ARV) national supply, which were imported centrally by the ministry of health, are dedicated for HIV treatment only [47]. The current antiretroviral therapy (ART) program in Indonesia does not give room for the use of ARV for PrEP use purpose [47,48]. Even to sell ARV that has been procured from an HIV program is impossible with the current regulation. Regardless of the legality, in Indonesia, in September 2020, a 30-day supply of PrEP regimen could be procured from in-country online platforms for 76–169 USD [20,21]. While, in 2018, in both Vietnam and Thailand, PrEP cost ranged around 15 USD [49,50]. In 2016, in Singapore, the branded PrEP Truvada could cost around 21 USD if purchased locally in Singapore. At the moment, PrEP in Indonesia is exclusive and pricey. This reason alone may limit potential users to access PrEP.

Some limitation to this study may include, first, as this study involved a relatively small sample of participants recruited from a single clinic, it may not be representative of all MSM/waria living in Bali. Second, the cross-sectional study design limited the study’s ability to assess changes in individuals over time. Third, social desirability bias from doing the survey with assistance instead of doing it personally in private, which might affect participants’ willingness to disclose risk behaviors or their interest in using PrEP.

In this study, we found that knowledge regarding PrEP among participants was originally low; however, once informed, interest in PrEP was high as expected. The structural barriers such as cost and procedures were somehow challenging in the Indonesian setting if PrEP is going to be implemented, thus, a PrEP program needs to address this challenge well. Participants with higher risks were more interested in PrEP use, which indicates an opportunity for targeted rollout of PrEP once PrEP programs do get started. In the light of this study findings, it is paramount that all HIV/AIDS-related stakeholders in Indonesia should continue the effort to increase sexual health and HIV-related knowledge among MSM/waria. Furthermore, if Indonesia has a goal of preventing the spread of HIV, it should give serious consideration to important issues such as how to set up a PrEP program, along with the HIV national guideline in a way that PrEP benefits are emphasized and promoted more to not discourage potential users. As such, we can expect that PrEP could contribute towards reaching zero new HIV cases in Indonesia.

## Figures and Tables

**Figure 1 tropicalmed-05-00158-f001:**
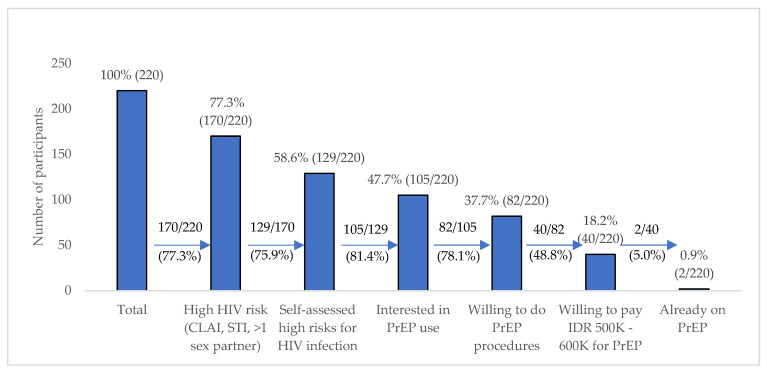
PrEP willingness for cascade 2 with the previous step as the denominator.

**Table 1 tropicalmed-05-00158-t001:** Socio-demographic, sexual, and HIV-related behavior in the last 6 months, and pre-exposure prophylaxis (PrEP) awareness/interest characteristic among men who have sex with men (MSM)/waria.

Variable	Number of Participants (n = 220)
**Demography**	
Age (years)	
<25	67 (30.5%)
25–29	66 (30.0%)
30–39	73 (33.2%)
40 and above	14 (6.4%)
Not married to a woman	200 (90.9%)
Education	
Primary education	26 (11.8%)
Senior high school	128 (58.2%)
University	66 (30.0%)
Full-time employee	163 (74.1%)
Income more than Bali minimum wage	105 (47.7%)
High social engagement as MSM/waria	68 (30.9%)
Adequate HIV risks/prevention knowledge	93 (42.3%)
**Sexual and HIV Prevention-Related Behavior in the Last 6 Months**	
Sex with women	73 (33.2%)
Multiple sex partner	111 (50.5%)
Sex with the regular partner/s	126 (57.3%)
Inconsistent condom use	136 (61.8%)
Sex with casual partner/s	128 (58.2%)
Paid for sex	34 (15.5%)
Sexually transmitted infection (STI) diagnoses	26 (11.8%)
Assessed HIV high risk	170 (77.3%)
Self-assessed HIV high risk	150 (68.2%)
**PrEP Awareness and Interest**	
Have heard of PrEP before	36 (16.4%)
Want to use PrEP	164 (74.5%)
Willing to do PrEP procedures	134 (60.9%)
Willing to pay 500K–600K IDR	84 (38.2%)
Have been in PrEP before	2 (0.9%)

**Table 2 tropicalmed-05-00158-t002:** Comparison of the participants with and without high HIV risk self-perception.

Variables	Self-Perception of no HIV Risk	Self-Perception of High HIV Risk	Crude Odds Ratio (OR)			Adjusted OR (aOR)		
	n = 41, n (%)	n = 129, n (%)	OR	95% CI	*p*-Value	aOR	95% CI	*p*-Value
Age								
<25 years	4 (9.8)	49 (38.0)	Ref	Ref	Ref			
25–29 years	10 (24.4)	45 (34.9)	0.37	0.11–1.25	0.110 *	0.30	0.11–1.30	0.0073
30–39 years	22 (53.7)	30 (23.3)	0.11	0.03–0.35	0.000 *	0.07	0.03–0.29	0.000 **
40 and above	5 (12.2)	5 (3.9)	0.08	0.02–0.41	0.002 *	0.06	0.01–0.34	0.002 **
Education								
Less than senior high school	8 (19.6)	15 (11.6)	Ref	Ref	Ref	Ref	Ref	Ref
Senior high school	23 (56.1)	72 (55.8)	1.67	0.63–4.44	0.304	1.43	0.44–4.69	0.552
University	10 (24.4)	42 (32.6)	2.24	0.74–6.74	0.151 *	2.51	0.68–9.26	0.167
Income more than Bali minimum wage	22 (53.7)	59 (45.7)	0.73	0.36–1.47	0.377			
High social engagement as MSM/waria	13 (31.7)	42 (32.6)	1.04	0.49–2.21	0.919			
Adequate HIV risk/prevention knowledge	15 (36.6)	61 (47.3)	1.55	0.75–3.21	0.232 *	2.10	0.89–4.95	0.088
Sex with women in the last 6 months	17 (41.5)	43 (33.3)	0.71	0.34–1.45	0.344			
>1 MSM/waria sex partner in the last 6 months	20 (48.8)	91 (70.5)	2.51	1.22–5.17	0.012 *	3.08	1.23–7.72	0.016 **
Inconsistent condom use in the last 6 months	36 (87.8)	93 (72.1)	0.36	0.13–2.30	0.047 *	0.57	0.18–1.83	0.348
Paid sex in the last 6 months	8 (19.5)	24 (18.6)	0.94	0.39–2.3	0.897			
STI diagnoses in the last 6 months	3 (7.3)	23 (17.8)	2.75	0.78–9.68	0.115 *	2.44	0.61–9.74	0.206

* *p*-value cut-off 0.25, ** *p*-value < 0.05.

**Table 3 tropicalmed-05-00158-t003:** Comparison of the participants with and without interest in PrEP use.

Variables	Not Interested In Prep Use	Interested in PrEP Use	Crude OR (OR)			Adjusted OR (aOR)		
	n = 24, n (%)	n = 105, n (%)	OR	95% CI	*p*-Value	aOR	95% CI	*p*-Value
Age								
<25 years	12 (50.0)	37 (35.2)	Ref	Ref	Ref	Ref	Ref	Ref
25–29 years	4 (16.7)	41 (39.1)	3.32	0.99–11.21	0.053 *	4.23	0.96–18.60	0.056
30–39 years	6 (25.0)	24 (22.9)	1.30	0.43–3.92	0.645	1.59	0.40–6.30	0.509
40 and above	2 (8.3)	3 (2.9)	0.49	0.07–3.27	0.458	1.57	0.15–16.95	0.707
Education								
Less than senior high school	1 (4.17)	14 (13.3)	Ref	Ref	Ref	Ref	Ref	Ref
Senior high school	14 (58.3)	58 (55.2)	0.30	0.04–2.44	0.258	0.31	0.03–3.23	0.330
University	9 (37.5)	33 (31.4)	0.26	0.03–2.27	0.224 *	0.26	0.02–3.41	0.303
Income more than Bali minimum wage	15 (62.5)	44 (41.9)	0.43	0.17–1.08	0.072 *	0.38	0.10–1.43	0.151
High social engagement as MSM/waria	15 (62.5)	27 (25.7)	0.21	0.08–0.53	0.001 *	0.17	0.06–0.51	0.002 **
Adequate HIV risk/prevention knowledge	13 (54.2)	48 (45.7)	0.71	0.29–1.74	0.455			
Sex with women in the last 6 months	5 (20.8)	38 (36.2)	2.16	0.74–6.24	0.157 *	1.37	0.38–4.88	0.628
>1 MSM/waria sex partner in the last 6 months	14 (58.3)	77 (73.3)	1.96	0.78–4.93	0.150 *	4.26	1.26–14.45	0.020 **
Inconsistent condom use in the last 6 months	14 (58.3)	79 (75.2)	2.17	0.86–5.47	0.100 *	4.80	1.43–16.09	0.011 **
Paid sex in the last 6 months	3 (12.5)	21 (20.0)	1.75	0.48–6.43	0.399			
STI diagnoses in the last 6 months	3 (12.5)	20 (19.1)	1.65	0.45–6.07	0.453			

* *p*-value cut-off 0.25, ** *p*-value < 0.05.

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
