# Peer review of "PrEP Use Awareness and Interest Cascade among MSM and Transgender Women Living in Bali, Indonesia"

_tropicalmed, 2020, doi:10.3390/tropicalmed5040158_

Round 1

Reviewer 1 Report

This work addresses a relevant topic and aims at exploring barriers for PrEP care package implementation among individuals at high risk of acquiring HIV in Bali. The cascade design and reporting of the survey is appropriate for a root cause analysis identifying crucial issues such as PrEP awareness. This allows to suggest interventions required to maximise PrEP uptake in such setting.  The information presented is of great interest to the relevant authorities that will commission such PrEP package of care in Bali. The authors are aware of the difficulties to extrapolate these results to the rest of Indonesia or other countries. This is well expressed in the limitations. The description of methods is clear and such methods seem adequate to identify gaps in the HIV prevention programmes. Other results also aid to identify target high-risk populations with a discordant self-perception of risk.  Hence, I consider this work valuable for publication.    

Author Response

We would like to thank for your positive comments on this manuscript.

We have submitted a revision of the paper based on suggestions from other referees.

All the best,

Reviewer 2 Report

The manuscript describes an important issue of raising awareness of the use pf PrEP among transgender women in Bali.

I have the following concerns that the authors should respond in order to improve the quality of the manuscript.

1- Line 54: Obtaining PrEP in Indonesia without a prescription is illegal and costly..."  the authors should describe the reasons of this in more details and if it is associated with health issues of the people.

2-Lines 83-84: the following lines should be discussed in discussion part and not in methods. (...Around the time of data collection period, there were limited PrEP campaigns in Indonesia, including in Bali. Therefore, we expected low PrEP awareness among participants....)

3-Line 147: ...Per months28": typing error ? what is 28?

4-Line 153: "(61.8%) participants reported inconsistent condom use..."

what are the consequences of this high number of participants ? 

5- Line 234: what kind of stigma ( religious , social ...etc)

finally I recommend acceptance of the manuscript after meeting the above inquires 

Author Response

Thank you for the comments provided, our responses are given in a point-by-point manner. The changing of  the manuscript is shown in red font colour.

The manuscript describes an important issue of raising awareness of the use pf PrEP among transgender women in Bali.

I have the following concerns that the authors should respond in order to improve the quality of the manuscript.

Response:

Thank you for your critical comments. We have revised the manuscript in accordance with your comments.

1- Line 54: Obtaining PrEP in Indonesia without a prescription is illegal and costly..."  the authors should describe the reasons of this in more details and if it is associated with health issues of the people.

Response:

It has been revised accordingly (line 61).

“Distribution of ARV as PrEP regulation is also not yet available. However, at the moment in Indonesia, PrEP can be bought online without prescription [20,21], despite the fact that ARV as used in PrEP can only be obtained through prescription [19].”

2-Lines 83-84: the following lines should be discussed in discussion part and not in methods. (...Around the time of data collection period, there were limited PrEP campaigns in Indonesia, including in Bali. Therefore, we expected low PrEP awareness among participants....)

Response:

It has been revised accordingly. We have moved the mentioned lines from the methods section to the discussion section.

Line – 98: “Regardless of participants prior knowledge on PrEP, all participants were provided with scripted information describing PrEP during the interview.”

Line – 243: “Around the time of data collection period, there were limited PrEP campaigns in Indonesia, including in Bali. Therefore, we expected low PrEP awareness among participants. To address this, all participants were provided with scripted information describing PrEP during the interview. Unsurprisingly, in this study, we identified an exceptionally low level of PrEP knowledge in the sample.”

3-Line 147: ...Per months28": typing error ? what is 28?

Response:

It was a typing error. It has been revised accordingly. It should be the reference number (in bracket). In fact, it should be [31] instead of [28]. The numbering transition was due to the expanded introduction section (as per suggestion from Reviewer 3) which include new references.

Line 165: “Three-quarters were employed full-time, and 47.7% reported their income to be more than the Bali provincial minimum wage of IDR 2.494 million per month [31].”

4-Line 153: "(61.8%) participants reported inconsistent condom use..."

what are the consequences of this high number of participants ?

Response:

More explanation has been added in the discussion session in accordance with your comment.

Line 270: “Moreover, if the 61.8% participants with inconsistent condom use kept on engaging in unsafe sex of CLAI, it could be challenging to keep the HIV status negative among them without any interventions.”

5- Line 234: what kind of stigma ( religious , social ...etc)

Response:

More explanation has been added.

Line 257: “One possible explanation could be that those with strong MSM/waria peer networks may be more likely to fear judgement or stigma from their peers about PrEP use or their sexual behavior, such as fear of rejection from partners (actual/potential), stigma of promiscuity and chemsex stereotypes labelling, and the fear of potential label stigma surrounding the PrEP medication and its users, as has been found in other setting [45].”

finally I recommend acceptance of the manuscript after meeting the above inquires

Response:

We would like to thank for your positive comments on this manuscript.

Reviewer 3 Report

The manuscript reports findings from a bio-behavioural survey of MSM and transgender women in Bali. The authors report the level of awareness of participants’ HIV risks and they corresponding knowledge about and interest in PrEP. I think this is a very important topic and is a clearly needed research in the area. At the same time, I have some comments about the data analysis and presentation that I think authors would need to address before potential acceptance.

Major comments

  • Regression analysis – how was your multivariable model built? Some of the variables that you decided to include were clearly poorly correlated with the outcome in the bivariate analysis (like education or HIV knowledge), why did you decide to keep them in by choosing a very high threshold for significance (0.25)? I suggest you re-run the multivariable analysis only with the variables that were significant at a more meaningful level (0.1?) Furthermore, have you checked for multicollinearity before putting all these variables together? Some of them will most likely be correlated (education/income/paid sex experience/etc).
  • I am a bit concerned about the construction of the outcome variable as the answers of what constitutes a positive or negative answer varied between variables (cascade steps 2 and 6, for example). I suggest a supplementary where you explain in more details how you constructed the variables.
  • The presentation of results is confusing. I suggest to delete table 2&3 and keep the diagram. The “%relative drop” in the last column of Table 2 is redundant. I do not see why the authors chose to present separately with/without PrEP awareness, I think it clogs the results. Again, I would keep the total (with awareness) in the main text, and if so desired moved the rest to a supplementary.

Minor comments

  • The Introduction should be expanded to include data from the neighbouring countries (or examples from other locations if regional data are limited) on the uptake of PrEP and any corresponding changes in the epidemics. Also, better understanding of the epidemic in Indonesia, not only in the MSM community, should be provided.
  • More information need to be provided on the recruitment. Were all clients who met the inclusion criteria invited? How did you ensure no double participation? Was the clinic you chose centrally located? What population is it more likely to capture?

[19] - please introduce the term waria in the abstract if you use it there

[43] – what is the prevalence in general population?

[58] – what is the cost?

[111] - you mention 6 steps?

Author Response

Response to Reviewer 3

The manuscript reports findings from a bio-behavioural survey of MSM and transgender women in Bali. The authors report the level of awareness of participants’ HIV risks and they corresponding knowledge about and interest in PrEP. I think this is a very important topic and is a clearly needed research in the area. At the same time, I have some comments about the data analysis and presentation that I think authors would need to address before potential acceptance.

Response:

Thank you for your critical comments. We have revised the manuscript in accordance with your comments.

Major comments:

Regression analysis – how was your multivariable model built? Some of the variables that you decided to include were clearly poorly correlated with the outcome in the bivariate analysis (like education or HIV knowledge), why did you decide to keep them in by choosing a very high threshold for significance (0.25)? I suggest you re-run the multivariable analysis only with the variables that were significant at a more meaningful level (0.1?) Furthermore, have you checked for multicollinearity before putting all these variables together? Some of them will most likely be correlated (education/income/paid sex experience/etc).

Response:

Before we run the multivariable model, we run bivariate regression analysis on each of independent variables mentioned in table 2 and table 3 (total 10 variables). For bivariate analysis, each of the independent variables was assessed individually in relation to its association with the variable of interest in both table 2 (self-perception of high HIV risk) and table 3 (interest in PrEP use). The variables which significance was <0.25 were included in the multivariate regression analysis. In the multivariate regression analysis, the independent variables which significance was <0.05 were reported as associated to the variable of interest in table 2 (self-perception of high HIV risk) and table 3 (interest in PrEP use). However, in the light of previous study that highlight the important of variables we used in the multivariate regression analysis, we have decided to stick on to the threshold of significance of <0.25.

We have conducted multicollinearity testing before running the regression analysis among independent variables and no highly correlated independent variables were detected among independent variables for both table 2 and 3.

Line 205: “From the multicollinearity testing we found that none of the independent variables was found to be highly correlated (r<0.5)”

Line 213: “From the multicollinearity testing we found that none of the independent variables was found to be highly correlated (r<0.5)”

I am a bit concerned about the construction of the outcome variable as the answers of what constitutes a positive or negative answer varied between variables (cascade steps 2 and 6, for example). I suggest a supplementary where you explain in more details how you constructed the variables.

Response:

We have revised the methods section in accordance with your comment, where we have added information on how we construct the variable in step 2 and step 6.

Line 110: “2.  Self-perceived high risk for HIV infection: Measured by the question, “Based on your sexual activities in the last 6 months, how likely do you perceive your risk of being infected with HIV?”; of 4 options provided for the participants (not at all at risk, less risky, risky, and highly risky) those answering “not at all” were classified as not perceiving themselves as high risk.”

Line 121: “6.  Willing to pay IDR 500,000 to 600,000 for PrEP (the cost of locally available PrEP at the time): Measured by the question, “How willing are you to use PrEP if it costs IDR 500,000-600,000 per month?”; of the 5 options provided for the participants (highly unwilling, unwilling, slightly willing, willing, and highly willing) those answering “highly unwilling” and “unwilling” were classified as unwilling.”

The presentation of results is confusing. I suggest to delete table 2&3 and keep the diagram. The “%relative drop” in the last column of Table 2 is redundant. I do not see why the authors chose to present separately with/without PrEP awareness, I think it clogs the results. Again, I would keep the total (with awareness) in the main text, and if so desired moved the rest to a supplementary.

Response:

We have moved table 2 and table 3 to the supplementary section and revised the result section in accordance with your comment.

Line 185: “In the first PrEP cascade (as shown in Table S1 as a supplementary) where PrEP awareness was included, three-quarters (n=170, 77.3%) were classified as at high HIV risk, and from these, 75.9% (n=129) perceived themselves to be at high risk. Of those with high risk self-perception, only 17 participants (13.2% of the previous step) were aware of PrEP, representing an 86.8% decrease from the previous step.”

Line 192: “In the second PrEP cascade (as shown in Figure 1 and Table S2 as a supplementary) where all participants were considered aware of PrEP, of the 129 with self-perceived high HIV risk, 81.4% (n=105) were interested in using PrEP. Of those interested, 78.1% (n=82) were willing to do the PrEP procedures, but only half (48.8%, n=40) of those willing to do the PrEP procedures were willing to pay IDR 500,000 to 600,000 per month for it. Two participants, representing 5.0% of those willing to pay and only 0.9% of all participants, were already taking PrEP. There were large percentage decreases (that is, of >10%) between each step of cascade 2.”

Minor comments

The Introduction should be expanded to include data from the neighbouring countries (or examples from other locations if regional data are limited) on the uptake of PrEP and any corresponding changes in the epidemics. Also, better understanding of the epidemic in Indonesia, not only in the MSM community, should be provided.

Response:

We have expanded the introduction along with the reference section in accordance with your comment.

Line 53: “Although plan to establish a pilot study on PrEP implementation for MSM population has been stated in its National Strategy and Action Plan 2015-2019 for HIV and AIDS prevention [13], Indonesia has not implemented PrEP and has fallen behind neighboring countries such as Thailand [14], where it has been integrated into its universal health coverage since 2018 [15] and result in an estimate of 16.000 – 17.000 PrEP users in Thailand as of July 9, 2020 [16]. Although literatures suggesting direct PrEP contribution towards HIV epidemic in Thailand is not yet readily available, the fact that HIV prevalence has slowed down in recent years in this country should not be overlooked [17].”

Line 39:“Based on the integrated HIV bio-behavioral surveillance (IBBS) report in 2018-2019, national HIV prevalence was 17.9%, 13.6%, 11.9%, and 2,1%  among men who have sex with men (MSM), people who inject drugs (PWID), transgender women/TGW (known locally as ‘waria’), and female sex workers respectively [2]. In Denpasar, Bali, Indonesia, the 2019 IBBS report showed that HIV prevalence among MSM was 38.1% [2], while the 2015 IBBS report showed that HIV prevalence among female (direct) sex workers was 4.8%, female (in-direct) sex workers was 5.6%, MSM was 36.0%, and incarcerated people 3.8%  [3].”

More information need to be provided on the recruitment. Were all clients who met the inclusion criteria invited? How did you ensure no double participation? Was the clinic you chose centrally located? What population is it more likely to capture?

Response:

We have revised the methods section in accordance with your suggestion.

Line 84: “We conducted a cross-sectional survey on HIV risk, PrEP awareness, and interest in PrEP among MSM/waria attending a non-government HIV testing and treatment clinic in Denpasar, Bali, Indonesia. The clinic is a very reputable clinic in its specialized services regarding sexually transmitted infections (STI) and HIV for the HIV key populations.”

Line 92: “All eligible participants were invited to participate in the study by clinic staff. After receiving study information and giving consent, participants were asked to complete an interviewer-administered survey. All participants were interviewed face to face by the first author only, therefore, double participation could easily be avoided.”

[19] - please introduce the term waria in the abstract if you use it there

Response:

It has been revised accordingly.

Line 20: “We aim to present a PrEP cascade among men who have sex with men (MSM) and transgender women (known locally as “waria”) in Denpasar, Bali from a cross-sectional survey with 220 HIV-negative MSM/waria recruited from one clinic in Denpasar.”

[43] – what is the prevalence in general population?

Response:

It has been revised accordingly.

Line 39: “In 2018, UNAIDS reported that there were about 640,000 people living with HIV (PLHIV) in Indonesia [1], with the prevalence of 0.4% among the general population [1]”

[58] – what is the cost?

Response:

It has been revised accordingly.

Line 68: “However, the emerging PrEP-related cost could highly likely prohibit most potential PrEP users to access PrEP abroad, as they must consider the fly-out transportation cost, in country accommodation and meal cost, as well as in country PrEP-related costs [22–25]”

[111] - you mention 6 steps?

Response:

We have added step 7 in the list of cascade steps in the methods section, in line with the following sentence in line 128.

Line 126: “7. Already in PrEP: Measured by the question, “Have you ever used ARV as PrEP before any sexual activities?”; those answering “yes” were classified as already on PrEP”